# Syncope in a Working-Age Population: Recurrence Risk and Related Risk Factors

**DOI:** 10.3390/jcm8020150

**Published:** 2019-01-29

**Authors:** Franca Barbic, Franca Dipaola, Giovanni Casazza, Marta Borella, Maura Minonzio, Monica Solbiati, Satish R. Raj, Robert Sheldon, James Quinn, Giorgio Costantino, Raffaello Furlan

**Affiliations:** 1Internal Medicine, Humanitas Clinical and Research Center, Biomedical Sciences Department, Humanitas University, 20089 Rozzano, Italy; franca.dipaola@humanitas.it (F.D.); maura.minonzio@humanitas.it (M.M.); raffaello.furlan@hunimed.eu (R.F.); 2Dipartimento di Scienze Biomediche e Cliniche "L. Sacco"-Università degli Studi di Milano, 20157 Milan, Italy; giovanni.casazza@unimi.it; 3Emergency Department, Ospedale di Rho, ASST Rhodense, 20017 Rho, Italy; martab79@virgilio.it; 4Fondazione IRCCS Ca’ Granda Ospedale Maggiore Policlinico, University of Milan, 20122 Milan, Italy; monica.solbiati@gmail.com (M.S.); giorgic2@gmail.com (G.C.); 5Libin Cardiovascular Institute of Alberta, Department of Cardiac Sciences, University of Calgary, Calgary, AB T2N 4Z6, Canada; satish.raj@ucalgary.ca (S.R.R.); sheldon@ucalgary.ca (R.S.); 6Division of Emergency Medicine, Stanford University, Stanford, CA 94305, USA; quinnj@stanford.edu

**Keywords:** syncope, working activity, syncope recurrence risk factors, high-risk job, syncope in working-age

## Abstract

Syncope in a worker undertaking risky tasks may result in fatalities for the individual or for third parties. We aimed at assessing the rate of syncope recurrence and the risk factors underlying the likelihood of syncope relapse in a working-age population. A prospective cohort of all patients aged 18–65 years consecutively admitted to the Emergency Department for syncope was enrolled. Risk of syncope relapse was assessed at a six-month, 1-year, and 5-year follow-up. Predictors of syncope recurrence have been evaluated at six months and 1 year from the syncope index by a multivariable logistic regression analysis. 348 patients were enrolled. Risk of syncope relapse was 9.2% at 6 months, 11.8% at 1 year, and 23.4% at 5 years. At 6-month follow-up, predictor of syncope recurrence was ≥3 prior lifetime syncope episodes. At 1-year, ≥3 prior lifetime syncope episodes, diabetes mellitus, and anaemia were risk factors for syncope relapse. There was an exceeding risk of recurrence in the first 6 months and a reduced risk of 3.5% per year after the first year. Anaemia, diabetes mellitus, and prior lifetime syncope burden are of importance when giving advice about the resumption of “high risk” jobs following a syncope episode.

## 1. Introduction

Syncope is a common clinical problem affecting about 6.2/1000 person-year [1]. It is usually associated with loss of postural tone resulting in falls [2,3]. Even clinically benign vasovagal episodes, which account for most of the syncope [1,4,5,6], may end in a fatal event for the individual or for third parties when occurring in hazardous conditions, such as driving coaches or trucks, working close to hot materials or flames, or handling chemicals. A recent paper [7] based on a Danish nationwide cohort of syncope patients aged 18–64 years found a 1.4-fold higher risk of work accidents and a 2-fold higher risk of occupation termination in syncope patients compared to the general working population. Of interest, the incidence rate ratio of work accidents was higher in individuals who suffered from recurrent syncope [7], highlighting the crucial role played by syncope relapse in that subset of patients.

To date, scientific society guidelines and expert consensus papers dealing with syncope clinical management have only partially addressed the relationships among syncope, its relapse, and work environments [8,9,10]. Few studies addressed the role of working activity on syncope occurrence and relapse [11,12,13,14,15].

While patients discharged from the Emergency Department (ED) usually receive clinical advice, the potential occupational risk is rarely addressed by physicians [8]. Recently, the need for an optimized assessment and management of patients with syncope before resuming a risky job has been emphasized by an Expert Consensus Statement [8].

There is some data on the risk of syncope recurrence in the general population [1,16,17,18], however there is no information on either the risk of syncope relapse or the recurrence risk factors specifically in a population suitable for work. These aspects are of paramount importance, as there may be age-dependent effects on the risk of syncope recurrence.

The present prospective study was aimed at assessing the syncope recurrence risk and the risk factors underlying the likelihood of syncope relapse in a group of patients aged 18–65 years. The evaluation of syncope recurrence risk is a crucial factor in computing the global risk of workers suffering from syncope [14].

## 2. Methods

### 2.1. Study Population

From a previously published cohort of 670 patients consecutively seen in the ED because of syncope [19], all 348 patients aged 18–65 years (“working age”) were enrolled (Figure 1). The patients were admitted to the ED of four hospitals in the Milan (Italy) area because of syncope having occurred within the previous 48 hours. The demographic and clinical features of the patients are reported in Table 1. Clinical follow-up was prospectively obtained at 6 months, 1 year, and 5 years. Details on the patient’s occupational status at the moment of the syncope were also collected. The study was approved by the Ethical Committee on Human Research of the Coordinating Centre (L. Sacco Hospital, University of Milan) in agreement with The Code of Ethics of the World Medical Association (no code number provided in accordance with the rules of times). Participants provided written informed consent to participate in the study in accordance with the Declaration of Helsinki.

### 2.2. Exclusion Criteria

Exclusion criteria encompass acute diseases diagnosed in the ED, including cases that required hospital admission irrespective of syncope, head injury reported before the loss of consciousness [10], concomitant diseases with a life expectancy of 6 months, documented alcohol or drug abuse, or lack of feasible follow-up (foreigners, homeless).

### 2.3. Study End Points

Three issues were addressed in the present study:The 6-month, 1-year, and 5-year risk of syncope recurrence after the index event.The risk factors potentially related to syncope relapse at 6 months and 1 year.The 6-month, 1-year, and 5-year prognosis after the index syncope.

### 2.4. Definitions

*Syncope recurrence*: additional syncope that occurred within 6 months, 1 year, and 5 years after the index event.

*Severe outcomes*: death, life-saving therapeutic procedures, including cardiopulmonary resuscitation, Pacemaker (PM) or Implantable Cardiac Defibrillator (ICD) implant, admittance to intensive care unit, acute anti-arrhythmic therapy.

*Abnormal ECG*: sinus tachycardia, atrial fibrillation, sinus pause ≥2 seconds, sinus bradycardia (heart rate between 35–45 beats per minute), conduction disturbances (e.g., bundle branch block, Mobitz II atrio-ventricular block), ECG signs of previous myocardial infarction or ventricular hypertrophy, multiple ventricular beats.

*Active workers*: individuals who were holding a job at the enrollment.

*Non-active workers*: individuals who were temporarily unemployed at the enrollment, including subjects who were early retirees but still fit to work.

### 2.5. Data Collection and Follow-Up

Six physicians obtained the ED reports to perform the initial screening of the patients. Subjects were evaluated prior to ED discharge or telephonically surveyed within 2 days regarding their occupational status, recurrence of syncope, and potential severe outcomes.

Six-month, 1-year, and 5-year follow-up data were collected by phone interviews. If patients were unreachable or unable to talk, their relatives or general practitioners were interviewed. All data on patient’s clinical history, presenting symptoms, physical examination findings, laboratory test results, and follow-up were collected by a physician at the coordinating centre and stored in a prospectively designed database.

### 2.6. Statistical Analysis

Continuous variables were reported as median and interquartile range (IQR). Categorical variables were reported as numbers and percentages.

Fisher exact test was used to compare demographic clinical features.

The Kaplan-Meier method was used to estimate the risk of syncope recurrence with its 95% confidence intervals (CI) at the available time points. Univariable and multivariable logistic regression models were fitted to assess the association between 6-month and 1-year syncope recurrence and the variables assessed at time of ED evaluation: age, gender, hospital admission, abnormal ECG, first syncopal episode, number of syncopal episodes over lifetime, trauma after syncope, presence of prodromal symptoms, cardiovascular disease, neoplasm, neurological disease, Chronic Obstructive Pulmonary Disease (COPD), diabetes, hypertension, and anaemia. A backward elimination procedure was used to find the best predictive model. The estimated odds ratios (OR) with their CIs were calculated. A *p* value lower than 0.05, 2-sided, was considered statistically significant. The statistical analyses were performed using the SAS statistical software (SAS Institute Inc. Cary, NC, USA; release 9.4).

## 3. Results

As shown in Figure 1A, 348 subjects aged 18–65 years were included. Six-month and 1-year follow-up were obtained in all 348 enrolled patients. Between 1-year and 5-year follow-up, 111 patients were lost from the study.

Demographic and clinical characteristics of the studied population are summarized in Table 1. In 41% of patients, there was no previous history of syncope. Ninety-one subjects (26%) reported ≥3 syncope events in their lifetime. Most were female and the large majority (85%) were discharged after ED evaluation, either because they were considered at low risk or because vasovagal was the likely origin of syncope. Indeed, more than two-thirds of enrolled patients did not suffer from comorbidities (64%). Most patients (76%) had syncope in an upright position, no trauma (80%), and suffered from pre-syncope symptoms. The ECG was normal at first evaluation in 84% of patients.

The studied population was subdivided into active and non-active workers (Table 1). Non-active workers were slightly older than active ones. Accordingly, hypertension, structural heart disease, and diabetes were more frequent in non-active than in active workers. Despite that, age and comorbidity differences had no effects on syncope recurrence, at 6-month, 1-year (Table 1), and 5-year follow-up. Therefore, no differences were observed in syncope recurrences between the two groups.

Syncope etiology was identified in 188 patients. Most syncope cases (*n* = 166) were vasovagal, and 7 were associated with anxiety or panic disorders. There were 5 syncope cases of cardiac origin, 2 PM, 2 ICD implants, and one patient received antiarrhythmic therapy.

### 3.1. Population Occupational Profile

Out of the 348 patients enrolled, 211 (61%) were active workers at the time of syncope. The remaining subjects, who were non-active workers at the time of enrollment, were not working because of temporary unemployment (*n* = 98) or early retirement (*n* = 37). All non-active workers were fit to work but two suffered from permanent disability. Activities carried out by active workers included white-collar and tertiary sector activities, manual labor (factory and building), and professional driving (Figure 1B). In 53 subjects (25% of the active workers), the index event occurred while working (Figure 1B).

Most of the employees (60%) went back to their previous job after the ED discharge. All of the subjects who fainted while working, including drivers and other people working in risky environments, returned to their job the day after the ED discharge without further evaluation.

### 3.2. Syncope Recurrences Time Course within 5-Years

Syncope recurrence within 5 years following the index event is shown by the Kaplan-Meier curve (Figure 2). The slope of the curve was steeper within the first 6 months. In that period, the risk of syncope recurrence was 9.2% (95% CI 6.6–12.8). In the second six-month period, the curve became flatter with a cumulative 11.8% (95% CI 8.8–15.7%) risk of recurrence within 1 year and 23.7% (95% CI 18.9–29.4%) within 5 years. The overall increase of recurrence risk was 3.5% per year, up to 5 years.

Out of all syncope recurrences, 44% occurred in the first 6 months from the index event. 51% of patients had only 1 syncopal recurrence in follow-up, whereas 28% of them suffered from ≥3 recurrences during the 5-year follow-up. Of interest, among patients admitted to ED for a first syncope event (40,5%), the rate of syncope recurrence was 8% (i.e., 11 out of 141; one subject suffered from diabetes mellitus and cardiovascular disease, one from anemia, one from hypertension) at 6 months. The rate of syncope recurrence was 2% (i.e., 3 out of 141; one patient suffered from diabetes mellitus) at 1 year.

### 3.3. Predictors of Syncope Recurrence at 6-Month and 1-Year Follow-Up

Predictors of syncope recurrence at a 6-month and 1-year follow-up are reported in Table 2. At 1-year follow-up, ≥3 lifetime syncope episodes, diabetes mellitus, and anaemia were predictors of syncope recurrence according to multivariate analysis (Table 2).

### 3.4. Six-Month, 1-Year, and 5-Year Prognosis

At 6-month follow-up, 8 severe outcomes were observed (1.7%).

At 1-year follow-up, an additional patient died, and 2 patients were readmitted to the hospital because of syncope.

In patients who had experienced syncope while working, 3 pacemakers (PM) were implanted. In this group, 7 patients reported a major trauma as a consequence of fainting.

Within 5 years from a syncope episode, 9% of patients suffered from severe outcomes.

The clinical features of the 111 subjects lost after the first year of follow-up were like those of the remaining 237 subjects, therefore, they had no effects on 5-year prognosis results.

## 4. Discussion

The main results of the current investigation are as follows:In total, 51.9% of patients presenting to the ED because of syncope were of working-age (18–65 years) and proved to be healthier compared to unselected syncope patients of previous studies (19–22).In total, 60.6% of the working age patients were active workers at that time and 25% of them fainted while working.The risk of syncope relapse was higher within the first 6 months, i.e., 9.2%, after the index event compared to the remaining follow-up periods (3.5% per year).Diabetes, anaemia, and ≥3 lifetime syncope episodes were risk factors independently associated with syncope recurrence at 1-year.

### 4.1. Syncope in A Working-Age Population

A recent investigation found a 1.4-fold higher risk of work accidents in subjects who suffered from recurrent syncope compared with the employed general population [7]. In addition, work accidents among syncope patients were more frequent in those working in manual labor sectors [7] that are often characterized by high risk.

A low clinical risk typified the majority of the patients of the present study. This was based on the high frequency of prodromes and the low rate of major trauma [9,10,20,21]. In addition, most of the patients (64.4%) were free from comorbidities, unlike in previous studies [19,22,23,24,25,26,27]. Accordingly, 85% of patients were discharged from the ED.

The prognosis of reflex syncope is benign. [1,10,28]. However, its consequences may be serious. Indeed, the sudden loss of postural tone and consequent falls may cause severe damage to the fainter [3,29,30] and third parties when occurring during high risk activities [7,13,14]. In keeping with this hypothesis are the Health and Safety at Work in Europe Reports, suggesting that most of fatal accidents occur as a consequence of actions characterized by “loss of control”, “slipping”, ”stumbling”, and “falling”. These work accident scenarios might be associated to syncope or presyncope.

Despite all these considerations, medical advice about when and how long to abstain from active work is often missed after discharge from the ED.

### 4.2. Risk of Syncope Recurrence in A Working-Age Population

The risk of syncope relapse in workers has previously been addressed by the Canadian Cardiovascular Society (CCS) Risk of Harm Formula for professional drivers [31,32] and by Barbic et al. for different working activities [14].

The risk of syncope recurrence is one of the relevant factors useful for assessing the global risk of harm to a worker, together with the time of exposure to a job task, the syncope facilitating features during work, and the estimated expected harm [14].

In the current study, there was a 9.2% risk of syncope recurrence within the first 6-months from the index event. Thereafter, the risk of relapse increased by 3.5% per year, up to 5 years. These data are consistent with the findings reported by Sumner et al. [33] at 1-year but seem different to what was reported in a recent meta-analysis [18] showing a linear time course of syncope recurrence with a rate of 22% at 2-year follow-up. Different age and clinical features of the patients may account for discrepancies.

The observation that syncope recurrence was similar in active and non-active workers points to a potential “intrinsic” risk of recurrence independent of the working activity. Specific working activity could increase the risk of syncope recurrence if syncope facilitating features during work are present [14].

### 4.3. Predictors of Syncope Recurrence

At 1 year, diabetes mellitus and anaemia were predictors of syncope recurrence in both active and non-active workers. The exact reasons explaining the association among syncope, syncope recurrence, diabetes, and anaemia are far from being completely understood. Hypotheses may be proposed as follows: Diabetes mellitus may be associated with various degrees of dysautonomia, in turn resulting in orthostatic hypotension and syncope [10]; anaemia is a potential cause of hypovolemia, a condition known to promote orthostatic intolerance and syncope [10]; the presence of ≥3 lifetime syncope episodes are an additional risk factor for syncope recurrence. This last finding is in keeping with previous studies that were characterized, however, by a different population mean age [33,34,35]. The number of lifetime syncope recurrence is in keeping with a likely underlying reflex mechanism, which, per se, tends to recur.

### 4.4. Limitations

The occupational features of our population, mostly characterized by white-collars, did not permit a robust analysis of the relationship between different types of working activities and syncope.

In addition, the present study was not designed to evaluate the role of syncope in determining work accidents. This would primarily require the identification and analysis of work accidents, searching thereafter for a potential relationship with syncope or pre-syncope.

In the current investigation, we excluded patients with loss of consciousness clearly identified as a consequence of head trauma, in accordance with the European Society of Cardiology Syncope Guidelines [10]. We acknowledge that some syncope episodes, particularly in blue-collar workers, might have been missed, because in the ED it may be difficult to obtain a detailed event history.

## 5. Conclusions

The working age population of the current study present an exceeding risk of syncope recurrence in the first 6 months and a reduced risk of 3.5% per year after the first year. Anaemia, diabetes mellitus, and prior lifetime syncope burden are of importance when giving advice about the resumption of “high risk” jobs following a syncope episode.

Syncope recurrence temporal distribution observed in the present study suggests that an observation period of 6–12 months before readmitting workers to a risky job might be a suitable time lag to reduce the risk of harm from a syncopal relapse, bearing in mind that the final aim is to reduce undue exclusion or absence from work and avoid an exceeding safety risk for patients and the community.

## Figures and Tables

**Figure 1 jcm-08-00150-f001:**
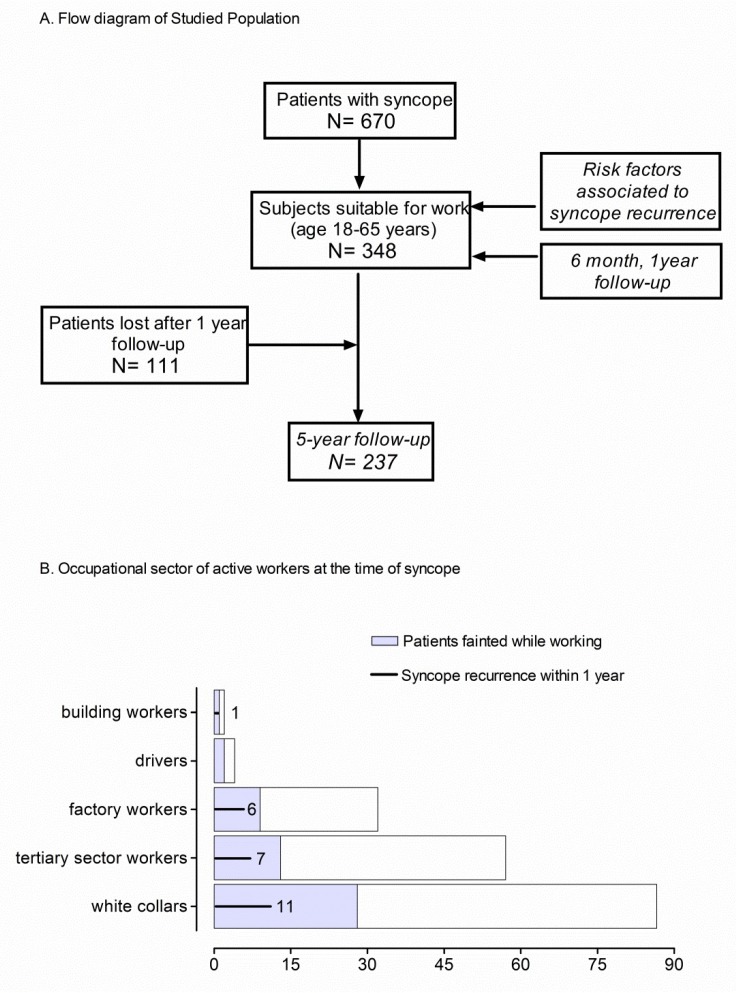
(**A**) The flow diagram shows the design of the study. Subjects suitable for work were enrolled as part of the STePS study (19). Recurrence risk of syncope relapse was estimated within 5-year follow-up. 111 patients were lost after the 1st year of follow-up. The risk factors associated with syncope relapse at 6-month and 1-year follow-up were evaluated in all patients. (**B**) Bars summarize the occupational sector of active workers at the time of first syncope. Each bar indicates the number of workers employed in specific occupational sector. Gray shaded areas indicate the number of subjects who fainted while working and black lines indicate the number of syncope recurrences within 1 year from the first syncope event in each different group of active workers.

**Figure 2 jcm-08-00150-f002:**
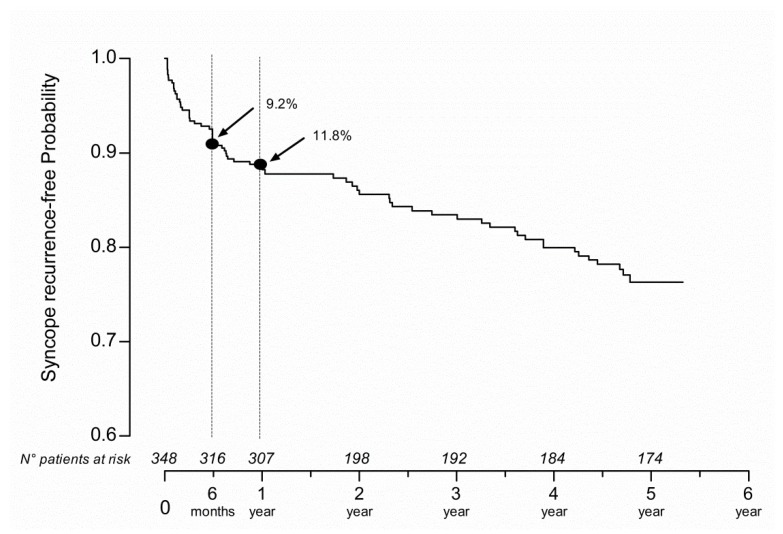
Risk of syncope recurrence within 5-year follow-up. Risk of syncope recurrence within 5-year follow-up as shown by the Kaplan-Meyer curve. On the X axis, the time of follow-up and the corresponding number of subjects at risk for syncope recurrence are reported. Notice that 9.2% and 11.8% of patients suffered from a syncope relapse within six-month and 1-year follow-up, respectively.

**Table 1 jcm-08-00150-t001:** Demographic and Clinical Features of the Studied Population.

		Active Workers	Non-Active Workers
Patients Enrolled	*n* = 348	*n* = 211	*n* = 137
Age, median “years” (IQR)	42 (29–54)	38 (28–48) *	52 (33–60)
Women, *n* (%)	199 (57.2)	126 (59.7)	73 (53.3)
Admitted, *n* (%)	52 (14.9)	26 (12.3)	26 (19)
Discharged *n*, (%)	296 (85.1)	185 (87.7)	111 (81)
Syncope recurrences, 6 m	32 (9.2)	20 (9.5)	12 (8.8)
Syncope recurrences, 1 year	41 (11.8)	24 (11.4)	17 (12.4)
**Past medical history, *n* (%)**			
≥3 lifetime syncope	91 (26.1)	63 (29.8)	28 (20.4)
No comorbidities	224 (64.4)	152 (72.0)	72 (52.6)
One or more comorbidities	124 (35.6)	59 (28.0)	65 (47.4)
Hypertension	66 (19.0)	23 (10.9)*	43 (31.4)
Structural heart disease	38 (10.9)	13 (6.2)*	25 (18.2)
Diabetes mellitus	23 (6.6)	7 (3.3)*	16 (11.7)
Chronic anaemia	17 (4.9)	12 (5.7)	5 (3.6)
Neurological disease	20 (5.7)	10 (4.7)	10 (7.3)
COPD	9 (2.6)	4 (1.9)	5 (3.6)
Cancer	7 (2.0)	4 (1.9)	3 (2.2)
Cerebrovascular disease	6 (1.7)	2 (0.9)	4 (2.9)
Ventricular Arrhythmias	4 (1.1)	1 (0.5)	3 (2.2)
Heart failure	3 (0.9)	0	3 (2.2)
**Index syncope history, *n* (%)**			
Supine/Sitting	73 (21.0)	45 (21.3)	28 (20.4)
Upright posture	266 (76.4)	161 (76.3)	105 (76.6)
During exercise	9 (2.6)	5 (2.4)	4 (2.9)
Trauma, *n* (%)	70 (20.1)	49 (23.2)	21 (15.3)
Abnormal ECG, *n* (%)	56 (16.1)	27 (12.8)	29 (21.2)
Absence of prodromes *n* (%)	71 (20.4)	40 (19.0)	31 (22.6)
First syncope, *n* (%)	141 (40.5)	93 (44.1)	48 (35.0)

Values expressed as *n* (%). IQR, interquartile range; m, months; COPD, Chronic Obstructive Pulmonary Disease; ECG, electrocardiogram. * *p* < 0.01 Active Workers vs Non-active Workers.

**Table 2 jcm-08-00150-t002:** Predictors of syncope recurrence within 6 months and 1 year from syncope index event.

	6-Months		1-Year			
	Univariable Analysis		Univariable Analysis		Multivariable Analysis	
	OR (95% CI)	*p*	OR (95% CI)	*p*	OR (95% CI)	*p*
Age > 40 “years”	1.84 (0.86–3.94)	0.12	1.49 (0.77–2.91)	0.24	-	
Gender	1.10 (0.53–2.31)	0.79	1.19 (0.61–2.33)	0.60	-	
No Hospital admission	2.82 (0.65–12.17)	0.17	1.71 (0.58–5.03)	0.33	-	
Abnormal ECG	1.53 (0.63–3.72)	0.35	2.14 (1.00–4.56)	0.05	-	
First syncopal episode	1.33 (0.62–2.86)	0.46	1.36 (0.69–2.7)	0.38	-	
Syncopal episodes ≥ 3	2.10 (0.99–4.42)	0.05	2.23 (1.14–4.38)	0.02	2.06 (1.03–4.10)	0.04
Trauma	1.85 (0.63–5.45)	0.27	1.54 (0.62–3.81)	0.35	-	
Absence of prodromal symptoms	1.43 (0.53–3.84)	0.48	1.28 (0.54–3.02)	0.57	-	
Cardiovascular Disease	1.74 (0.67–4.51)	0.25	1.25 (0.49–3.18)	0.64	-	
Neoplasm	1.67 (0.19–14.29)	0.64	1.25 (0.15–10.69)	0.84	-	
Neurological Disease	2.68 (0.84–8.57)	0.10	1.97 (0.62–6.20)	0.25	-	
COPD	2.95 (0.59–14.81)	0.19	2.2 (0.44–10.96)	0.34	-	
Diabetes	1.53 (0.43–5.46)	0.51	2.93 (1.08–7.91)	0.03	2.85 (1.04–7.83)	0.04
Hypertension	1.22 (0.50–2.95)	0.66	1.23 (0.56–2.73)	0.60	-	
Anemia	2.68 (0.84–8.57)	0.10	3.59 (1.30–9.94)	0.01	3.51 (1.25–9.83)	0.02

OR, Odds Ratio; CI, Confidence Interval; other abbreviations as in Table 1.

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
