# Peer review of "Syncope in a Working-Age Population: Recurrence Risk and Related Risk Factors"

_jcm, 2019, doi:10.3390/jcm8020150_

Round 1
Reviewer 1 Report
The author chose the working-age patients who visited emergency department due to syncope and collected the followed-up data at 6 months, 1 year and 5 years after syncope. The rate of syncope recurrence and the risk factors for the syncope relapse had been determined. The rate of recurrence with the first 6 months was higher than the later times points. ≥3 prior lifetime syncope events, diabetes mellitus and anemia are the risk factor for syncope relapse. However, some description or explanation is not clear.
Specific comments:
1. Page 2 line 46, “Only few studies have focused on that issue [11-15].” Please clarify which issue is not well known.
2. Page 5 line 133. “Ninety-one subjects (26%) reported ≥3 syncope events in their lifetime.” However, table 1 did not show any date about the patient number who have ≥3 syncope events before.
3. There is an inconsistency on the percentage of the active worker who had syncope while working. Page 5 line 153 showed 25%. However, page 7 line 194 showed 30%. Please double check and correct it.
4. Please marked the number for each area on Figure 1 or switch it to table. It is very confused to read the number.
5. It is very confused about the percentage which marked on the curve on Figure 2. At 6 months, (348-322)/348x100%=7.5%. Please clarify how you got 9.2%.
6. It is also very interesting to know the rate of syncope recurrence and the risk factors for the syncope relapse on the first syncope population (40.5%).
Author Response
See file attached.

Reviewer 2 Report
The authors have investigated that the recurrence risk and related risk factors of syncope in working-age population in Italy. They demonstrated that the reasonable recurrence rate of syncope and several risk factors associated with syncope recurrence in their study population. This study seems to be worth for publication because it contains important information about syncope in working population.
Major points:
The authors demonstrated that diabetes mellitus (DM) and anemia were risk factors for syncope recurrence at 1 year. Please describe the reason or speculation for why DM and anemia do associate with syncope recurrence in discussion.
Their conclusions are too long and not attractive. Please omit the sentences from line 251 to 254 and from line 257 to 358, and insert the sentences of line 26-29 in abstract: There was an exceeding risk of recurrence in the first 6 months (9.2%) and a reduced risk of 3.5% per year after the first year in working-age population. Anemia, DM, and prior lifetime syncope burden are of particular importance when giving advice about the resumption of “high risk” jobs following a syncope episode.
Minor points:
Page 5, line 133-134: The authors mentioned “91 subjects (26%) reported ≥3 syncope events in their lifetime.” Please shows this issue in Table 1.
Page 7, line 186: Does “serious outcomes” mean same as “severe outcomes”?
In Table 1, index syncope history of “supine/sitting” should be indicated separately because usually vasovagal syncope could not occur in the supine position but it can occur in the sitting position.
Author Response
See file attached
